# Changes in Estimated Body Composition and Physical Fitness of Adolescent Boys after One Year of Soccer Training

**DOI:** 10.3390/children10020391

**Published:** 2023-02-16

**Authors:** Cíntia França, Diogo V. Martinho, Élvio Rúbio Gouveia, Francisco Martins, Adilson Marques, Tiago Ribeiro, Marcelo de Maio Nascimento, Helder Lopes, Ana Rodrigues, Andreas Ihle

**Affiliations:** 1Department of Physical Education and Sport, University of Madeira, 9020-105 Funchal, Portugal; 2LARSYS, Interactive Technologies Institute, 9020-105 Funchal, Portugal; 3Research Center in Sports Sciences, Health Sciences, and Human Development (CIDESD), 5000-801 Vila Real, Portugal; 4University of Coimbra, Research Unit for Sport and Physical Activity (CIDAF), Faculty of Sport Sciences and Physical Education, 3004-504 Coimbra, Portugal; 5Center for the Interdisciplinary Study of Gerontology and Vulnerability, University of Geneva, 1205 Geneva, Switzerland; 6Faculty of Human Kinetics, University of Lisbon, 1499-002 Lisbon, Portugal; 7CIPER, Faculty of Human Kinetics, University of Lisbon, 1495-751 Lisbon, Portugal; 8ISAMB, Faculty of Medicine, University of Lisbon, 1649-020 Lisbon, Portugal; 9Department of Physical Education, Federal University of Vale do São Francisco, Petrolina 56304-917, Brazil; 10Department of Psychology, University of Geneva, 1205 Geneva, Switzerland; 11Swiss National Centre of Competence in Research LIVES—Overcoming Vulnerability: Life Course Perspectives, 1015 Lausanne, Switzerland

**Keywords:** flexibility, strength, body composition, football, adolescents

## Abstract

Sports participation is one of the most popular forms of physical activity among youngsters. This study aimed to examine the changes in the estimated body composition, strength, and flexibility of adolescent boys after 12 months of soccer training compared with those of age-matched controls with non-sports participation. We assessed 137 boys (62 soccer players and 75 controls) at baseline (TM1) and 12 months later (TM2). The differences in estimated body composition, strength, and flexibility were investigated using a repeated measure analysis of variance. The analysis revealed a significant main effect of soccer training on fat mass (F = 73.503, *p* ≤ 0.01, η2 = 0.59) and fat-free mass (F = 39.123, *p* ≤ 0.01, η2 = 0.48). Over time, the soccer group decreased their fat mass and increased their fat-free mass, while the opposite results were observed for the controls. Among physical fitness tests, a substantial effect of soccer training was evidenced for the sit-up performance (F = 16.224, *p* ≤ 0.01, η2 = 0.32). Regarding the time factor, significant effects were noted for height and handgrip strength. No significant differences were detected for flexibility. Overall, the benefits of soccer training were exhibited by the larger improvements in fat mass, fat-free mass, sit-ups, and handgrip strength performance, underlining the important role of soccer participation during adolescence.

## 1. Introduction

Sports participation is one of the most popular forms of physical activity (PA) among children and adolescents [1,2]. Consistently, the literature has described the benefits of sports participation in youth for both mental and physical health [3,4,5]. However, global trends show a decline in overall PA levels with increasing age in adolescence, which has been a subject of great concern to health entities [5]. Among youngsters, soccer is one of the most attractive sports environments [1], contributing to worldwide research development. In Portugal, demographic statistics in 2020 pointed to nearly 191,000 players enrolled in organized soccer competitions, about one-third of the total number of Portuguese federated practitioners [6].

Overall, extensive studies in youth regarding topics such as functional capacities [7,8,9], training methods efficacy [10,11], and talent identification [12,13] have been conducted. Meanwhile, the health-related benefits of recreational soccer participation have also been reported [14,15]. Improvements in aerobic capacity, muscle mass, and body fat have been observed in participants from several age groups [16,17], from both sexes [18], and with different health statuses [19,20] following a programmed soccer intervention. Recreational soccer led to significant cardiovascular and muscular enhancements in untrained men throughout a 12-week training period [16]. Another study design to examine the effects of 16 weeks of soccer training in women reported favorable outcomes for cardiovascular and body fat profiles [18]. Furthermore, studies in youth reported that recreational soccer enhanced estimated aerobic fitness, 20 m sprint performance, and horizontal jumping performance in untrained healthy male adolescents [21]. Moreover, a 12-week recreational soccer intervention also proved to reduce body mass index (BMI), waist circumference, and fat mass percentage (FM%) among obese and nonobese boys and girls aged between 12 and 17 years [22].

Although data on youth soccer are extensive, few authors have aimed to assess the benefits of regular soccer training participation in competitive clubs. Indeed, two studies compared the differences between soccer players and age-matched controls after 8 months [23] and 2 years of monitoring [24] in Tunisian and Finnish adolescents, respectively. As expected, in terms of body composition, in a sample of Tunisian boys aged 14.5 ± 0.4 years, the authors reported that soccer players were significantly taller, heavier, and had lower FM% than controls at baseline and follow-up. Additionally, soccer players outperformed their peers in speed (10 and 30 m linear sprint) and in estimated VO2 max (predicted by the Yo-Yo intermittent recovery test level-1) [23]. A study on Finnish boys between 10 and 15 years concluded that soccer players performed better in speed, agility, and aerobic and anaerobic capacity in the age groups examined than the age-matched controls [24]. In the previously mentioned studies, no significant differences were found between groups in strength, mainly assessed through jumping ability. The literature describes physical fitness variables as crucial to game performance. Furthermore, body composition parameters have been consistently assessed in a sports context, particularly to avoid the detrimental effect of body fat in game actions such as sprinting, jumping, and changing direction [2].

Although body composition data are available from past investigations regarding physical fitness variables, an analysis of the literature indicates a lack of details concerning static strength, abdominal strength, endurance, and flexibility. Therefore, to close these important gaps, the aim of this study was to examine the changes in the estimated body composition, strength, and flexibility of boys after 12 months of soccer training compared to age-matched controls (non-sports participants). It was hypothesized that decreases in FM% and gains in fat-free mass, strength, endurance, and flexibility would be significantly greater among soccer players compared with age-matched controls.

## 2. Materials and Methods

### 2.1. Participants and Procedures

One hundred and thirty-seven boys were recruited to participate in this study. The optimal sample size was calculated using G*Power 3.1 (Düsseldorf, Germany), and a priori repeated measures analysis of variance (ANOVA) indicated a total sample size of 110 participants to achieve 85% power to detect an interaction effect of 0.25 at the 0.05 level of significance.

A total of 62 boys were soccer players (age = 15.3 ± 1.4 years, height = 168.1 ± 7.0 cm, body mass = 63.6 ± 14.8 kg, training experience of at least two years) from an initial sample of 102 players (Figure 1). All players competed at the regional level, with an average of 4 weekly training sessions of 75 min each, plus on1e match at the end of the week between the end of September and the beginning of June (regional or local championship that qualifies a team for the final national championship phase). For the soccer group, participants had to meet the following inclusion criteria: (i) aged between 14 and 16 years, (ii) registered in competitive soccer at the Portuguese Federation, and (iii) affiliated with the local Soccer Association in the last two seasons. This study only included players with a minimum of 85% of the total training sessions performed in their club. Training frequency was checked through the data collected daily by their coaches.

In addition, 75 boys were included in the control group (age = 15.4 ± 1.8 years, height = 166.1 ± 7.9 cm, body mass = 61.5 ± 15.1 kg), recruited from a total sample of 172 boys evaluated in the local schools. In the control group, participants met the following inclusion criteria: (i) aged between 14 and 16 years, and (ii) no sports participation besides their school Physical Education classes for the last two years. Participants in the control group participated in the research project entitled “Physical Education in Schools from the Autonomous Region of Madeira” EFERAM-CIT; https://eferamcit.wixsite.com/eferamcit (accessed on 15 October 2022).

The current study only considered participants with two moments of evaluation (baseline (TM1) and follow-up (TM2)). From the initial sample, 40 soccer players were excluded due to age or lack of assessment at least once. Additionally, 97 boys were excluded from the control group due to age, the lack of evaluation at one time point, or the engagement in a sports activity between TM1 and TM2.

Anthropometry, static strength, abdominal muscular strength and endurance, flexibility, and upper-body strength were evaluated twice (TM1 and TM2). All test assessments were performed by trained staff from the research team, with knowledge of each protocol. A general warm-up routine of about 15 min combining slow jogging, dynamic movements (i.e., high knees and skips), and static and dynamic stretching was performed before the fitness assessment. The procedures used in this study were approved by the Ethics Committee of the Faculty of Human Kinetics, CEIFMH Nº34/2021, and followed the Declaration of Helsinki. All participants were volunteers, and informed consent was obtained from their legal guardians.

### 2.2. Anthropometry

Height was measured using a portable stadiometer (SECA 213, Hamburg, Germany) to the nearest 0.1 cm. Body mass was assessed using a portable scale (SECA 760, Hamburg, Germany) to the nearest 0.1 kg. The waist circumference assessment was performed just above the iliac crest using a nonelastic measurement tape (SECA 201, Hamburg, Germany) to the nearest 0.1 cm. The participants were in standing position with arms hanging freely. Skinfold thickness was measured to the nearest 0.1 mm at five sites (biceps, triceps, subscapular, abdominal, and calf) using a skinfold caliper (Harpenden Skinfold Caliper, West Sussex, Chichester, UK). All measurements were taken following the International Society for the Advancement of Kinanthropometry (ISAK) guidelines [25] by experienced investigators. Nine trainee physical education teachers collected anthropometric data. Prior training concerning the protocols was provided by the university training program, which included theoretical and practical sessions. A pilot study was conducted to evaluate 15 adolescents aged between 16 and 18 years. These adolescents were assessed twice with an interval of one week. The results indicated acceptable to good test–retest interobserver reliabilities and good to excellent test–retest intraobserver reliabilities.

Fat mass percentage (FM%) was estimated through the Slaughter equation for men, using triceps and calf skinfolds [26]: PFDWB = 0.735 (triceps + calf) + 1.0.

### 2.3. Muscular Strength

The handgrip was used to evaluate static strength [27]. The protocol included three alternated trials for each arm using a hand dynamometer (Jamar Plus+, Chicago, IL, USA). Participants were standing and asked to hold a dynamometer in one hand, laterally to their trunk, with the elbow at a 90° position. Participants squeezed the dynamometer as hard as possible from this position for about two seconds. If the dynamometer touched the participant’s body, the assessment was repeated. Before data collection, one experimental trial was performed with each hand to guarantee the protocol’s execution. The rest interval between trials was 60 s, and the best score of the three trials performed with the participant’s dominant side was retained for analysis.

The sit-up protocol consisted of performing the highest number of repetitions in 30 s to assess abdominal muscular strength and endurance [28]. Participants started in a sitting position, torso vertical, hand behind their neck, and knees bent at 90°, and feet flat on the floor. Then, participants stretched out on their backs with their shoulders touching the floor and straightened up to the sitting position immediately after. Their elbows had to contact their knees; otherwise, the repetition was not counted. Before data collection, participants were asked to perform five experimental repetitions to ensure correct execution. The total number of repetitions performed during the protocol corresponded to the test score.

### 2.4. Flexibility

A variation of the sit and reach test was used to assess flexibility [29]. The protocol used a sit-and-reach trunk flexibility box (32.4 cm high and 53.3 cm long) with a 23 cm heel line mark. Participants sat barefoot in front of the box, with one knee fully extended and the heel placed against the box. The other knee was bent with the longitudinal arc of the foot placed against the fully extended knee [30]. Then, participants were asked to put their hands on top of each other, palms down, and push forward for the measuring cycle. The forward position was repeated twice, and the third forward stretch was held for 3 s, corresponding to the test score. The test was conducted unilaterally (right and left legs). Participants performed one experimental trial for each side to ensure correct execution. During data collection, two trials were performed, and the best score was used for analysis.

### 2.5. Statistics

Descriptive statistics are presented as means ± standard deviations. All data were checked for normality using the Kolmogorov–Smirnov test. The repeated measures ANOVA was conducted to test the effect of soccer training (group), intraindividual changes over 12 months (time), and the interaction term (group × time). The effect size for each factor (12 months participation, type of sport, and interaction) is given by eta squared, quantitatively interpreted as follows [31]: η2 ˂ 0.1 (trivial), 0.1 ≤ η2 ˂ 0.3 (small), 0.3 ≤ η2 ˂ 0.5 (moderate), 0.5 ≤ η2 ˂ 0.7 (large), 0.7 ≤ η2 ˂ 0.9 (very large), and 0.9 ≤ η2 (nearly perfect). In addition, Cohen d-values were calculated [32] and interpreted as follows [31]: d ˂ 0.2 (trivial), 0.2 ≤ d < 0.6 (small), 0.6 ≤ d < 1.2 (moderate), 1.2 ≤ d < 2.0 (large), 2.0 ≤ d < 4.0 (very large), and d ≥ 4.0 (nearly perfect). All analyses were performed using IBM SPSS Statistics software 28.0 (SPSS Inc., Chicago, IL, USA) and GraphPad Prism (version 5.00 for Windows, GraphPad Software, San Diego, California USA, www.graphpad.com (accessed on 10 October 2022). The significance level was set at 5%.

## 3. Results

Table 1 summarizes the descriptive statistics for each group regarding the estimated whole-body composition and physical fitness tests at baseline (TM1) and follow-up (TM2). At baseline, the soccer players were significantly taller and presented lower FM% and fat-free mass than the controls. Moreover, at baseline, the soccer players performed substantially better in the sit-up and flexibility tests.

Table 2 presents the results of repeated measures ANOVA to examine the effect of sports participation (soccer vs. control), time (TM1 vs. TM2), and the interaction term between sport and time. The analysis revealed a significant main group effect on estimated body composition, particularly in FM% (F = 73.503, *p* ≤ 0.01, η2 = 0.59) and waist circumference (F = 9.371, *p* ≤ 0.01, η2 = 0.25). Among the physical fitness tests, a substantial group effect was observed for sit-ups (F = 16.224, *p* ≤ 0.01, η2 = 0.32) but not for the handgrip strength and flexibility tests. Regarding the time effect, very large effects on estimated body composition and strength assessment were found. The greatest effect sizes were identified for height (F = 132.12, *p* ≤ 0.01, η2 = 0.70) and handgrip strength (F = 91.986, *p* ≤ 0.01, η2 = 0.64). Finally, the analysis showed a significant interaction effect (group × time) in FM% (F = 30.822, *p* ≤ 0.01, η2 = 0.43), sit-ups (F = 13.057, *p* ≤ 0.01, η2 = 0.30), and handgrip strength (F = 4.298, *p* ≤ 0.01, η2 = 0.18).

Figure 2 shows the intraindividual changes (TM1 and TM2) for fat mass, fat-free mass, sit-ups, and handgrip strength for the soccer players and controls. The results for FM% showed that the control group tended to gain fat mass (d = 0.50), while the fat mass of soccer players reduced over one year. Consistent results were obtained for fat-free mass and small changes in the control group for sit-ups (d = 0.45) and handgrip strength (d = 0.32). Moderate changes over one year were observed in the soccer players.

## 4. Discussion

This study examined the changes in the estimated body composition, strength, and flexibility of youth boys after 12 months of soccer training compared with those of age-matched controls with non-sports participation. As expected, the results of the current study indicated significant changes in the estimated body composition and strength performance after 12 months. However, larger improvements were observed for soccer players in FM%, fat-free mass, sit-ups, and handgrip strength performance.

The sample in the present study was composed of adolescent boys aged 15.3 ± 1.4 years, so increases in height and body mass should reflect the influence of growth [2]. At baseline, the soccer players were taller and heavier than the controls, which is consistent with the results found in elite Tunisian adolescent males. The mean values indicated that the soccer players were nearly 7 cm taller and almost 7 kg heavier than the controls at baseline [23]. However, in Finnish boys, regional soccer players were above the median reference line for height and body mass according to Finnish growth charts. Still, they were shorter and lighter than age-matched control groups [24]. These contradictory results might illustrate the players’ selection process. A previous study found that soccer players from the under-14 age category selected to play in the under-15 age category were taller and heavier than non-selected youngsters [31]. Longitudinal data among boys showed height mean values that maintained their position relative to reference values over time, suggesting that height is not influenced by systematic sports participation [32].

In the present study, a substantial decrease in estimated FM% (−0.9%) and an increase in fat-free mass (+2.8 kg) were observed in the soccer group between TM1 and TM2. Due to growth and biological maturation, these changes in the estimated body composition would be expected over time in boys during adolescence [2]. However, the results of the control group were the opposite, showing a significant increase in FM% (+4.5%) and a slight decrease in fat-free mass (−1.9 kg) between assessments. Although these results are substantially different, the same trend was observed in Tunisian male adolescents. After 8 months, soccer players showed a decrease of nearly 1.5% in FM%, while controls presented a decline of 1.8% [23]. The literature underlines that soccer training is associated with benefits to body composition, contributing to the decrease in FM% and increase in fat-free mass [33]. In youth, the aerobic demands of soccer average between 70 and 90% of their maximum heart rate, leading to high-fat loss compared with other sports activities with lower intensity levels [34]. The frequent execution of high-intensity actions during a soccer game, such as sprinting, jumping, and dribbling, should contribute to higher levels of fat-free mass in soccer players [35,36].

A significant time effect in both groups for sit-ups and handgrip strength performance was found in the present study. These results were expected because the sample was framed into middle adolescence, where gains in muscle mass and strength are continuous in boys [37]. Indeed, at age 14, boys become significantly stronger for nearly all tested muscle groups [38]. Of note, the mean scores in the soccer group at follow-up (sit-ups: +4 repetitions; handgrip strength: +1 kg) were significantly better than those of the control group (sit-ups: +2 repetitions; handgrip strength: 0 kg). Regardless, these results should be interpreted with caution due to the possible confounder effect of maturity status. In a sample of 78 youth Belgian soccer players aged 15–16 years, the morphological measures of the more mature players were higher. They performed significantly better in all fitness assessments than their late-to-mature peers. However, the authors reported that soccer-specific and nonspecific motor coordination tests (i.e., dribble test without the ball and dribble test with the ball) were not related to maturity status [39]. Another study conducted among male paddlers indicated significantly superior fitness, anthropometric attributes, and better race times in advanced mature youngsters [40].

Concerning flexibility, soccer players significantly outperformed their control peers at TM1. However, no overall changes were seen in the sit-and-reach performance over time due to soccer training, while the controls increased their baseline performance. Indeed, the lower scores showed by the controls at baseline would allow them a greater opportunity to improve their performance over time. However, the literature mentions that boy’s flexibility capacity tends to decline up to mid-adolescence [37,41]. Previous research on youth soccer players at different age ranges described the progressive decrease in the lower-extremity range of motion following advancing age [42]. Systematic stretching training was proven to increase the range of motion during childhood and adolescence [43]. Because flexibility is a component of physical fitness and players’ athletic development [44], it should be included as part of the training process, particularly during the critical early stages of soccer players’ long-term development.

The literature consistently encourages sports participation among children and adolescents [37]. The benefits of soccer training for estimated body composition and strength, compared with non-sports involvement, were evident in the results of the present study. However, considering other physical fitness components, such as power, speed, and aerobic and anaerobic capacity, would allow a more in-depth analysis of group differences. In addition, the participants’ maturity status and training details, such as previous training experience, training load, and structure of training sessions, were not considered, representing the current study’s limitations. Despite the findings underlining the positive influence of soccer training in male adolescents, particularly in decreasing estimated FM% and enhancing fat-free mass, sit-ups, and handgrip strength performance, future research, including more details concerning players’ training and competition structure, is still needed.

## 5. Conclusions

The results of the present study indicate the benefits of soccer training for estimated fat mass and fat-free mass and physical fitness performance in male adolescents. After 12 months, the soccer group showed significantly decreased levels of estimated FM% and enhanced fat-free mass. In contrast, FM% tended to increase, and a slight decrease in fat-free mass was observed in the age-matched controls. Soccer players significantly outperformed their control peers in the sit-ups and handgrip strength performance. Although changes in body composition and physical performance levels are expected during adolescence due to biological maturation, particularly at the age ranges assessed in this study, the findings underline the positive effect of soccer training during youth from a healthy perspective.

## Figures and Tables

**Figure 1 children-10-00391-f001:**
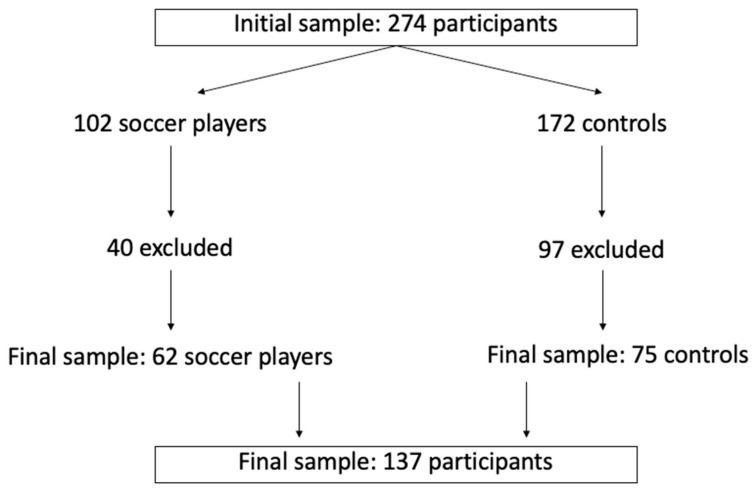
Flowchart presenting participants’ selection.

**Figure 2 children-10-00391-f002:**
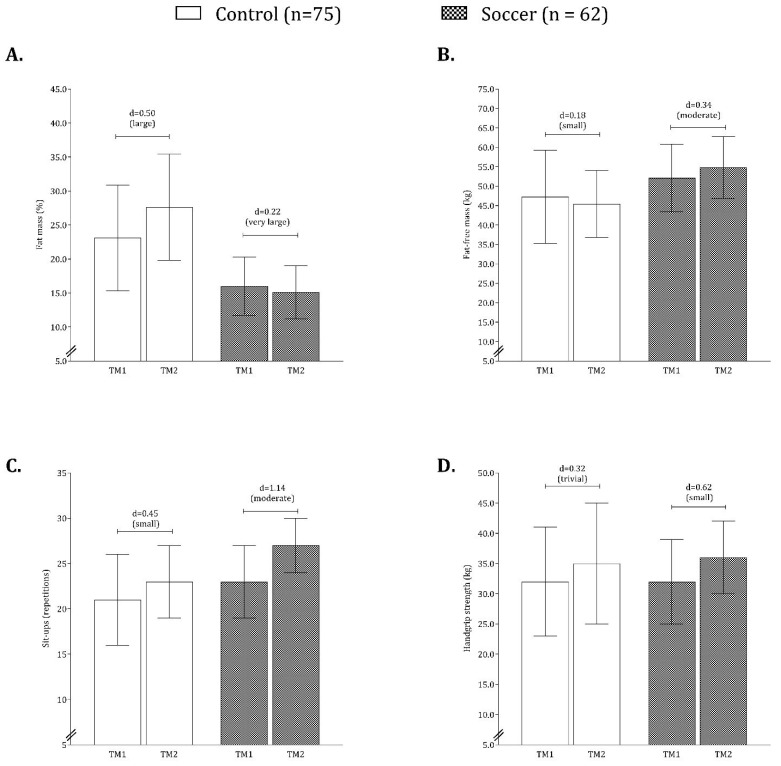
Intraindividual changes (TM1 and TM2) between groups for fat mass (**A**), fat-free mass (**B**), sit-ups (**C**), and handgrip strength (**D**).

**Table 1 children-10-00391-t001:** Descriptive statistics for each group at baseline and follow-up, and comparison between groups at baseline.

Dependent Variable	Groups						
	Control (n = 75)		Soccer (n = 62)				
	Baseline	Follow-Up	Baseline	Follow-Up	Groups Comparison at Baseline		
	Mean ± Standard Deviation				Mean Difference (95% CI)	t	*p*
Chronological age (years)	15.4 ± 1.8	16.4 ± 1.8	15.3 ± 1.4	16.3 ± 1.4	0.10 (−0.44 to 0.65)	0.378	0.71
Height (cm)	166.3 ± 7.7	168.4 ± 6.9	169.9 ± 9.1	173.2 ± 7.8	−3.57 (−6.41 to −0.72)	−2.483	≤0.01 **
Body mass (kg)	61.7 ± 15.0	63.9 ± 14.8	62.0 ± 10.2	64.6 ± 9.0	−0.30 (−4.58 to 3.98)	−0.139	0.89
Waist circumference (cm)	80.5 ± 13.1	75.4 ± 5.6	80.7 ± 12.8	75.2 ± 5.1	5.25 (1.92 to 8.58)	3.128	≤0.01 **
Fat mass (%)	23.1 ± 7.8	27.6 ± 10.1	16.0 ± 4.3	15.1 ± 3.9	7.06 (4.96 to 9.16)	6.661	≤0.01 **
Fat mass (kg)	14.4 ± 6.5	18.5 ± 10.0	9.9 ± 3.3	9.7 ± 2.8	4.48 (0.87 to 2.76)	5.164	≤0.01 **
Fat-free mass (kg)	47.3 ± 12.0	45.4 ± 8.6	52.1 ± 8.7	54.9 ± 8.0	−4.78 (−8.39 to −1.17)	−2.618	≤0.01 **
Sit-ups (n)	21 ± 5	23 ± 4	23 ± 5	27 ± 3	−1.64 (−3.08 to −0.21)	−2.268	0.03 *
Handgrip strength (kg)	32 ± 9	32 ± 7	35 ± 10	36 ± 6	−0.20 (−2.90 to 2.50)	−0.147	0.88
Sit and reach right (cm)	25 ± 9	31 ± 7	32 ± 7	31 ± 7	−6.03 (−8.77 to −3.28)	−4.334	≤0.01 **
Sit and reach left (cm)	24 ± 9	31 ± 7	31 ± 6	31 ± 6	−7.14 (−10.00 to −4.28)	−4.941	≤0.01 **

** p* ≤ 0.05, *** p* ≤ 0.01.

**Table 2 children-10-00391-t002:** Results of repeated measures ANOVA to examine the effects of sports participation, 12-month follow-up, and interaction among male adolescents (controls, n = 75; soccer players, n = 62).

Dependent Variable	Effects								
	Group Effect (Control vs. Soccer)			Time Effect			Interaction (Group × Time)		
	F	*p*	η^2^	F	*p*	η^2^	F	*p*	η^2^
	**Factorial ANOVA**								
Chronological age (years)	0.140	0.71	0.01						
Height (cm)	10.806	0.01 *	0.27	132.12	≤0.01 **	0.70	7.114	≤0.01 **	0.22
Body mass (kg)	0.126	0.73	0.03	40.32	≤0.01 **	0.48	0.264	0.61	0.04
Waist circumference ≤ e (cm)	9.371	≤0.01 **	0.25	0.001	0.99	0.01	0.343	0.56	0.05
Fat mass (%)	73.503	≤0.01 **	0.59	13.403	≤0.01 **	0.30	30.822	≤0.01 **	0.43
Fat mass (kg)	39.123	≤0.01 **	0.48	21.431	≤0.01 **	0.37	26.630	≤0.01 **	0.41
Fat-free mass (kg)	20.142	≤0.01 **	0.36	0.929	0.337	0.08	27.303	≤0.01 **	0.41
Sit-ups (n)	16.224	≤0.01 **	0.32	90.325	≤0.01 **	0.63	13.057	≤0.01 **	0.30
Handgrip strength (kg)	0.514	0.47	0.07	91.986	≤0.01 **	0.64	4.298	0.04 *	0.18
Sit and reach right (cm)	0.351	0.55	0.05	0.601	0.45	0.06	0.845	0.36	0.05
Sit and reach left (cm)	0.822	0.37	0.07	0.763	0.38	0.07	0.763	0.39	0.07

** p* ≤ 0.05, *** p* ≤ 0.01.

## Data Availability

The data presented in this study are available upon request from the corresponding author.

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
