# Peer review of "Changes in Estimated Body Composition and Physical Fitness of Adolescent Boys after One Year of Soccer Training"

_children, 2023, doi:10.3390/children10020391_

Round 1

Reviewer 1 Report

Thanks for inviting me to review this interesting paper that examined the effect of soccer training on body composition and physical fitness among a group of middle adolescents. Generally, the introduction and discussion are well-rounded. The only concerns I may have when reading the article are most related to the methods and results enlisted below. 

1. I am curious how much the soccer group has been trained for the soccer games and whether the control group is also trained for some other sports. The differences in group backgrounds should be clarified for the good of data interpretation. 

2. I assume the follow-period should be the same for soccer and control groups. However, why is there a discrepancy in the between-group age difference between TM1 and TM2 In Table 1?    

3. Further, I found that trends in body composition change are similar between the soccer and control groups. The between-group difference in body composition was obvious at baseline. That between-group difference remained unchanged over the study period because both groups lost their fat mass as they aged. 

4. There are some interesting findings in physical fitness that should be addressed.  As expected, the results found sit-ups improved and handgrip strength unchanged in the soccer group. On the other hand, sit-and-reach was more improved in the control group, as explained by the authors in the discussion. I just wonder why the finding was not statistically significant.  

Author Response

  1. I am curious how much the soccer group has been trained for the soccer games and whether the control group is also trained for some other sports. The differences in group backgrounds should be clarified for the good of data interpretation.

Response 1: We appreciate the reviewer’s comment. The authors’ tried to clarify better this information in the “Participants and Procedures” section (lines 101-116).

  1. I assume the follow-period should be the same for soccer and control groups. However, why is there a discrepancy in the between-group age difference between TM1 and TM2 In Table 1?

Response 2: We thank the reviewer's detailed analysis. In Table 1, the age of the control group was not corrected at the follow-up. The authors really apologize for this mistake. We have now corrected the data.

Reviewer 2 Report

Dear authors, The work that you present, despite having a good line, must be improved and enriched at a methodological level, as well as taking into account many more aspects that can condition the results, such as the type of training, the structure of the sessions, the volume load used in the development of each content or physical capacity etc... I strongly encourage you to delve into this line and provide much more robust data that supports the conclusions that you can contribute in this field.

Author Response

Dear authors, The work that you present, despite having a good line, must be improved and enriched at a methodological level, as well as taking into account many more aspects that can condition the results, such as the type of training, the structure of the sessions, the volume load used in the development of each content or physical capacity etc... I strongly encourage you to delve into this line and provide much more robust data that supports the conclusions that you can contribute in this field.

Response 1: We appreciate the reviewer’s overall feedback, and we also believe that providing more details concerning the training conditions would be far more informative to the analysis of physical fitness. Unfortunately, we do not have these details available which limited our methodological approach. The authors will consider including this information in future data collection. Even though, we believe that this study brings important insights concerning the topic of sports participation during adolescence and reinforces the positive effects of soccer training in this age range.

Reviewer 3 Report

The main idea of the investigation is not very original and it does not add a lot of information to the area of the body composition, maturity and sport. However, the design and the discussion are appropriate and it could be intersting for readers

Some issues must be addressed before publication

1) Indicate the previous training experience of the soccer players in years

2) The sedentary group, did they do PE? Did they do other non-structured physical activity?

3) Include more limititations to the study. (i.e. the training experience of previous years)

4) The manuscript must be reviewed by a native speaker and adapt to an academic language

- I.E. "our study...." Avoid personal forms.

5) Explain better the relationship between body composition, physical fitness and biological maturity

6) Include these references for that purpose

Vandendriessche, J. B., Vaeyens, R., Vandorpe, B., Lenoir, M., Lefevre, J., & Philippaerts, R. M. (2012). Biological maturation, morphology, fitness, and motor coordination as part of a selection strategy in the search for international youth soccer players (age 15–16 years). Journal of sports sciences, 30(15), 1695-1703.

López-Plaza, D., Alacid, F., Muyor, J. M., & López-Miñarro, P. Á. (2017). Sprint kayaking and canoeing performance prediction based on the relationship between maturity status, anthropometry and physical fitness in young elite paddlers. Journal of sports sciences, 35(11), 1083-1090

Author Response

1) The main idea of the investigation is not very original and it does not add a lot of information to the area of the body composition, maturity and sport. However, the design and the discussion are appropriate and it could be interesting for readers.

Response 1: We appreciate the reviewer’s overall positive feedback and the opportunity to revise this paper. We have now updated the manuscript considering the reviewer’s feedback and feel that it has improved its overall quality.

2) Indicate the previous training experience of the soccer players in years

Response 2: To meet the inclusion criteria, soccer players should have at least two years of previous training experience. This information was added in Participants and Procedures section (line 102).

3) The sedentary group, did they do PE? Did they do other non-structured physical activity?

Response 3: The control group was only participating in PE sessions without any other type of sports participation. The authors’ tried to clarify this information better in the Participants and Procedures section (line 115).

4) Include more limititations to the study. (i.e. the training experience of previous years)

Response 4: As recommended, the authors have included the training experience of previous years as the limitations of the present study. Also, the author’s introduced the lack of other training details (such as training load and structure of training sessions) as limitations since we recognize the importance of these variables in providing a more detailed analysis (line 298-300).

5) The manuscript must be reviewed by a native speaker and adapt to an academic language. I.E. "our study...." Avoid personal forms.

Response 5: We appreciate the reviewer's feedback concerning the writing. We have performed a deep review of the text and improved the manuscript's overall writing.

6) Explain better the relationship between body composition, physical fitness and biological maturity. Include these references for that purpose.

Response 6: The authors followed the reviewer’s advice and introduce the recommended references to better explain the relationship between body composition, physical fitness, and biological maturation in the Discussion section (lines 273-280).

Reviewer 4 Report

Dear authors,
Congratulations on your article. It is very interesting and well written.
The introduction is perfectly structured and with novel quotations. Congratulations. However, there is a lack of information about the importance of body composition, strength and flexibility and why you focus on these parameters. The inclusion of this information would be appreciated.
Please introduce the hypothesis of your study.
The method is generally adequate. However, you are required to include information about the sample's weekly training and competition volume range.
A flow chart would be appreciated to better visualise the evolution of the sample.
Please include information about the sample size calculation of the present study. What error is assumed with the sample included in the present study?
Did you do any familiarisation with the fitness tests? How?
Please include the make and model of the treadmill used for the measurement of girths. 
Please include information about the training of the anthropometry measurers and your intra- and inter-rater METs.
Please indicate with which hand you made the handgrip.
Why did you choose the sit-up and which muscles are assessed for strength and muscular endurance with this test? The choice of the test is questionable to say the least.
Why did you perform the sit and reach unilaterally? This test when done unilaterally is not called sit and reach. Please give it the correct name.
Did you perform any kind of warm-up before the fitness tests? If so, what was it, and what was the basis for your choice of warm-up?
Statistical analysis is adequate.
The results are well presented. Congratulations. However, why is it that although the protocol states that the sit and reach was also done bilaterally, the data is not in the results?
The discussion is quite adequate. However, the limitations of the study need to be further explored.
In the conclusions "benefits for body composition" is too generic and does not cover the number of anthropometric variables measured, which go beyond body composition. Please qualify this in the conclusion and coin a more specific term when referring to body composition throughout the manuscript (title, objectives...).

Author Response

  1. Congratulations on your article. It is very interesting and well written. The introduction is perfectly structured and with novel quotations. Congratulations. However, there is a lack of information about the importance of body composition, strength and flexibility and why you focus on these parameters. The inclusion of this information would be appreciated.

Response 1: The authors appreciate the reviewer’s overall positive feedback. The authors added this information at the end of the Introduction section (lines 86-88).

  1. Please introduce the hypothesis of your study.

Response 2: As recommended, the authors have introduced the study’s hypothesis (lines 91-93).

  1. The method is generally adequate. However, you are required to include information about the sample's weekly training and competition volume range.

Response 3: As recommended, the authors tried to clarify this information in the Participants and Procedures section (lines 103-106).

  1. A flow chart would be appreciated to better visualise the evolution of the sample.

Response 4: The layout was done according to the reviewer’s suggestion (see Figure 1).

Please include information about the sample size calculation of the present study. What error is assumed with the sample included in the present study?

Response 5: The optimal sample size was calculated using G*Power 3.1. A priori repeated measures ANOVA indicated a total sample size of 110 participants to achieve 85% power to detect an interaction effect of 0.25 at the 0.05 level of significance. This information was added at the beginning of the Participants and Procedures section (lines 98-100).

  1. Did you do any familiarisation with the fitness tests? How?

Response 6: Details concerning the familiarization with each fitness test were added in the Methods section for each protocol.

  1. Please include the make and model of the treadmill used for the measurement of girths.

Response 7: As recommended, the authors added more details to the text (line 144-145).

  1. Please include information about the training of the anthropometry measurers and your intra- and inter-rater METs.

Response 8: Nine trainee Physical Education teachers collected anthropometric data in schools. Prior training concerning the protocols was provided by the University training program, which included theoretical and practical sessions. A pilot study was conducted to evaluate 15 adolescents aged between 16 and 18 years. These adolescents were assessed twice with an interval of one week. Results indicated acceptable to good test-retest inter-observer reliabilities and good to excellent test-retest intra-observer reliabilities. This information was included in the Anthropometry section (lines 151 to 156).

  1. Please indicate with which hand you made the handgrip.

Response 9: The handgrip was performed with both hands; however, the authors only used the results of the dominant limb for analysis. This information is considered in the text.

  1. Why did you choose the sit-up and which muscles are assessed for strength and muscular endurance with this test? The choice of the test is questionable to say the least.

Response 10: We thank the reviewer’s pertinent observation. This sit-ups protocol is referred to as useful to assess abdominal strength and muscular endurance in past investigations, such as:

Esco, M. R., Olson, M. S., & Williford, H. (2008). Relationship of push-ups and sit-ups tests to selected anthropometric variables and performance results: A multiple regression study. The Journal of Strength & Conditioning Research, 22(6), 1862-1868.

Tomkinson, G. R., Carver, K. D., Atkinson, F., Daniell, N. D., Lewis, L. K., Fitzgerald, J. S., ... & Ortega, F. B. (2018). European normative values for physical fitness in children and adolescents aged 9–17 years: results from 2 779 165 Eurofit performances representing 30 countries. British journal of sports medicine, 52(22), 1445-1456.

The authors have rewritten the text to provide a clearer reader’s experience, referring to “abdominal muscular strength and endurance” instead of simply “muscular strength and endurance”.

  1. Why did you perform the sit and reach unilaterally? This test when done unilaterally is not called sit and reach. Please give it the correct name.

Response 11: The authors thank the reviewer’s important feedback on this matter. We have updated the methods section concerning flexibility assessment according to past references.

  1. Did you perform any kind of warm-up before the fitness tests? If so, what was it, and what was the basis for your choice of warm-up?

Response 12: A general warm-up routine of about 15 minutes combining jogging, dynamic movements (i.e. high knees and skips), and static and dynamic stretching was performed before the fitness assessment. This information was included at the end of the Participants and Procedures section and followed previous guidelines, such as:

Ramos, S., Volossovitch, A., Ferreira, A. P., Fragoso, I., & Massuça, L. (2019). Differences in maturity, morphological and physical attributes between players selected to the primary and secondary teams of a Portuguese Basketball elite academy. Journal of sports sciences, 37(15), 1681-1689.

Murtagh, C. F., Brownlee, T. E., Rienzi, E., Roquero, S., Moreno, S., Huertas, G., ... & Erskine, R. M. (2020). The genetic profile of elite youth soccer players and its association with power and speed depends on maturity status. PloS one, 15(6).

  1. Statistical analysis is adequate. The results are well presented. Congratulations. However, why is it that although the protocol states that the sit and reach was also done bilaterally, the data is not in the results?

Response 13: We have performed both assessments (unilateral and bilateral), but we only used the unilateral data in this study. We have removed the bilateral state in the protocol section to better clarify the readers, and we appreciate the reviewer’s detailed revision.

  1. The discussion is quite adequate. However, the limitations of the study need to be further explored.

Response 14: The authors added more limitations to the study, such as the lack of training and competition information (i.e., training load, and structure of training sessions) (lines 298-300).

  1. In the conclusions "benefits for body composition" is too generic and does not cover the number of anthropometric variables measured, which go beyond body composition. Please qualify this in the conclusion and coin a more specific term when referring to body composition throughout the manuscript (title, objectives...).

Response 15: Following the reviewer’s suggestion we have updated the term “body composition” to “estimated body composition” throughout the text.

Reviewer 5 Report

The authors had compared indicators of body composition and selected motor characteristics in young soccer players aged 15.3 and 16.3 years and in a control group aged 15.4 and 16.0 years. The gap between the initial and follow-up examination for the soccer players lasted one year, but only 0.6 years for the control group. The authors should explain this discrepancy in the study design.

The most significant changes were found in body composition, the percentage of body fat in controls during 0.6 years decreased from 28.3% to 16.0%, i.e. to 56% of the initial value, and in soccer players decreased in one year from 27.6% to15.1%, i.e. to 55% of the original value. These changes seem extreme and would require a detailed explanation.

The values listed in Table 1 show inaccuracies, if the fat mass in the initial examination corresponded to 14.4 kg, it corresponded to 23.4% of the total body weight of 61.5 kg, not the indicated 28.3%. Similarly, the sum of fat mass and fat-free mass at the baseline examination in soccer players (45.4 + 18.8 kg) corresponds to a total body weight of 64.2 kg and does not match the specified value 63.6 kg; or, fat mass 18,8 kg does not correspond to 27,6 % but 29,6 %.

In general, the majority of the results in boys selected into soccer training were better both at baseline examination and also at the end of follow up, and, therefore, it is difficult to attribute them only to the effect of sports training. As the authors themselves state in the discussion of the limitations of the study, other important components of physical activity were not monitored, nor was the biological development of boys in both monitored groups taken into account. 

Author Response

  1. The authors had compared indicators of body composition and selected motor characteristics in young soccer players aged 15.3 and 16.3 years and in a control group aged 15.4 and 16.0 years. The gap between the initial and follow-up examination for the soccer players lasted one year, but only 0.6 years for the control group. The authors should explain this discrepancy in the study design.
    Response 1: We thank the reviewer's detailed analysis. In Table 1, the age of the control group was not corrected at the follow-up, and the authors really apologize for this mistake.
  2. The most significant changes were found in body composition, the percentage of body fat in controls during 0.6 years decreased from 28.3% to 16.0%, i.e. to 56% of the initial value, and in soccer players decreased in one year from 27.6% to15.1%, i.e. to 55% of the original value. These changes seem extreme and would require a detailed explanation.

Response 2: The authors assume and apologize for some errors in the data presented in Table 1, which is now corrected. Both groups were evaluated after one year. We also tried to use the reviewer’s feedback to improve the Discussion section.

  1. The values listed in Table 1 show inaccuracies, if the fat mass in the initial examination corresponded to 14.4 kg, it corresponded to 23.4% of the total body weight of 61.5 kg, not the indicated 28.3%. Similarly, the sum of fat mass and fat-free mass at the baseline examination in soccer players (45.4 + 18.8 kg) corresponds to a total body weight of 64.2 kg and does not match the specified value 63.6 kg; or, fat mass 18,8 kg does not correspond to 27,6 % but 29,6 %.

Response 3: The authors agree with the reviewer’s commentary and have now revised Table 1, the following results (including Figure 2), and further discussion.

  1. In general, the majority of the results in boys selected into soccer training were better both at baseline examination and also at the end of follow up, and, therefore, it is difficult to attribute them only to the effect of sports training. As the authors themselves state in the discussion of the limitations of the study, other important components of physical activity were not monitored, nor was the biological development of boys in both monitored groups taken into account. 

Response 4: We agreed with the reviewer’s perspective, and we recognize the study’s limitations, which we aim to address in future studies.

Round 2

Reviewer 2 Report

Dear Author,

Despite the fact that the text has been significantly improved and enriched with new data, my opinion remains the same, to reject the document, since at a methodological level too many variables that intervene and can interfere with the results obtained have not been taken into account.

 Greetings

Author Response

Dear Author,

Despite the fact that the text has been significantly improved and enriched with new data, my opinion remains the same, to reject the document, since at a methodological level too many variables that intervene and can interfere with the results obtained have not been taken into account. Greetings

Response 1: We appreciate the reviewer's feedback and respect his decision. We recognize the manuscript's limitations, which will consider in future research.

Reviewer 5 Report

Some inconsistencies still persist in the area of the basic data shown in Table 1. In the second column, the sum of FFM 52.1 kg and FM 18.5 kg does not correspond to the indicated body weight of 63.9 kg, but to the weight of 70.6 kg. Similarly, in the third column, the sum of FFM 45.4 kg and FM 9.9 kg does not correspond to the indicated body weight of 62.0 kg, but to the weight of 55.3 kg. The data listed in Table 1 must be corrected and then the athors should use the corrected data in the discussion and in the conclusions.

Author Response

Some inconsistencies still persist in the area of the basic data shown in Table 1. In the second column, the sum of FFM 52.1 kg and FM 18.5 kg does not correspond to the indicated body weight of 63.9 kg, but to the weight of 70.6 kg. Similarly, in the third column, the sum of FFM 45.4 kg and FM 9.9 kg does not correspond to the indicated body weight of 62.0 kg, but to the weight of 55.3 kg. The data listed in Table 1 must be corrected and then the athors should use the corrected data in the discussion and in the conclusions.

Response 1: The reviewer is correct. We apologize for the mistake. The values are now corrected.